# Symbiosis-Inspired Knowledge Distillation for Incremental Object Detection

Mingyue Zeng [1]   De Cheng[✉] [1]   Zhipeng Xu [1]   Huaijie Wang [2]   Nannan Wang [1]   Xinbo Gao [2]

## Abstract

Incremental object detection (IOD) aims to extend detectors to new categories while retaining previously acquired knowledge. Existing methods often adopt a class incremental learning perspective, separating feature spaces to sharpen decision boundaries. However, this separation-oriented paradigm may overlook object symbiosis in detection, where co-occurrence and occlusion introduce spatial and semantic dependencies that benefit from shared representations. Ignoring these dependencies distorts the shared representations, exacerbates confusion between old and new classes, and accelerates catastrophic forgetting. To address this, we propose Symbiosis-Inspired Knowledge Distillation (SIKD), which explicitly leverages object symbiosis at two complementary levels. Spatial Symbiosis Distillation (SpSD) focuses on symbiotic regions where the old model responds with high overlap to objects in the new task. It preserves generalizable old class cues, suppresses class-specific bias and redundancy, and distills the refined evidence to the new model at matched spatial locations with slot-aligned supervision. Semantic Symbiosis Distillation (SeSD) maintains class level structure by forming confidence weighted prototypes for old classes and aligning their inter class soft ranks over the old class logits, which stabilizes the semantic topology during adaptation. Extensive experiments demonstrate the effectiveness and superiority of the proposed method.

## 1. Introduction

Object detection has advanced from two-stage frameworks (Girshick, 2015; Ren et al., 2016) to efficient one-

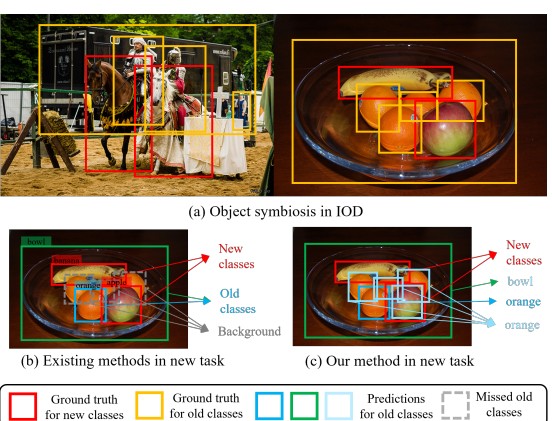

(a) Object symbiosis in IOD

(b) Existing methods in new task     (c) Our method in new task

*Figure 1.* Illustration of (a) object symbiosis in IOD, (b) existing methods in new task, and (c) our method in new task. In (b) and (c), arrows indicate how regional features are classified. In (c), the new-task class *apple* shares coarse features with the old class *orange* while also retaining class-specific cues.

stage detectors (Ge et al., 2021; Tian et al., 2019) and end-to-end transformer architectures (Zhu et al., 2020; Liu et al., 2024). Large-scale pretraining (Dai et al., 2021) and stronger benchmarks have further improved accuracy. However, in real deployments (Cheng et al., 2024; Xu et al., 2025), label spaces evolve as new categories appear. Retraining from scratch for every update is costly and often infeasible when prior data cannot be stored or accessed due to privacy or licensing. Incremental object detection (IOD) addresses this setting by learning new categories while preserving knowledge of learned ones using only the annotations available at each task.

Unlike class incremental classification (Zhou et al., 2024a; Masana et al., 2022), where each task provides complete labels for its classes, IOD trains on images that may still contain old objects while only the current categories are annotated. This mismatch pushes unlabeled old-class objects to be treated as background or drift toward new-class labels during training, which accelerates catastrophic forgetting.

To address this issue, existing IOD methods (Liu et al., 2023b; Kang et al., 2023; Mo et al., 2024; Kim et al., 2024; Wang et al., 2025c; Zhang et al., 2025a; Wang et al., 2025d) rely on the old model to mine old-class signals in the current data. As illustrated in Fig. 1(b), they retain only high-

[1]State Key Laboratory of Integrated Services Networks, School of Telecommunications Engineering, Xidian University, Xi'an, China [2]School of Electronic Engineering, Xidian University, Xi'an, China. Correspondence to: De Cheng <dcheng@xidian.edu.cn>.

*Proceedings of the 43rd International Conference on Machine Learning*, Seoul, South Korea. PMLR 306, 2026. Copyright 2026 by the author(s).

confidence old-class detections with low Intersection over Union (IoU) to new-class ground truth. These "clean" predictions are treated as the only source of old-task knowledge. This design follows a classification mindset that separates old and new features to reduce entanglement and sharpen decision boundaries (Rebuffi et al., 2017; Li et al., 2024). However, it overlooks the essential property of object symbiosis in detection. As shown in Fig. 1(a), objects naturally co-occur in shared contexts (e.g., *orange* and *apple* as fruits) and occlude one another (e.g., *person* riding *horse*), which create spatial and semantic dependencies that call for a unified feature space. Filtering supervision to only "clean" cases erodes symbiosis-bearing signals and biases the model toward the newly annotated categories, which increases old–new confusion and forgetting.

Instead, we propose Symbiosis-Inspired Knowledge Distillation (SIKD), a framework that maintains a unified feature space by leveraging object symbiosis across both spatial and semantic dimensions. As shown in Fig. 1(c), the old model processing new-task images reveals two symbiotic patterns. Unseen objects are mapped to semantically similar old classes, while partially visible old objects retain detection despite occlusion. We treat these as symbiotic regions encoding shared knowledge rather than noise. By preserving the consistent feature patterns presented in these regions, the model sustains a unified feature space across incremental tasks.

Concretely, SIKD distills symbiotic cues at two levels: instance-level spatial structure and class-level semantic topology, through Spatial Symbiosis Distillation (SpSD) and Semantic Symbiosis Distillation (SeSD). SpSD focuses on co-occurrence and occlusion regions, where it applies a Consistent Feature Enhancement (CFE) module to stabilize overlap-heavy features by reinforcing transferable patterns and suppressing spurious old-class activations. The enhanced features are then distilled to the new model via slot-aligned supervision to preserve spatial dependencies. In parallel, SeSD constructs confidence-weighted prototypes from both symbiotic and non-symbiotic regions and preserves their relative ordering in the old-class subspace via soft rank alignment, thereby maintaining the old-class semantic structure during incremental updates. Together, these two components improve knowledge retention across incremental steps. Our contributions are summarized as follows:

- We reinterpret IOD through object symbiosis (co-occurrence and occlusion), exposing the limits of classification-style feature separation in detection.

- We propose Symbiosis-Inspired Knowledge Distillation (SIKD), with Spatial Symbiosis Distillation (SpSD) to preserve spatial dependencies in symbiotic regions and Semantic Symbiosis Distillation (SeSD) to preserve old-class semantic topology.

- Extensive experiments achieve state-of-the-art performance, and ablations and visual analyses support our method.

## 2. Related Work

### 2.1. Incremental Learning

Incremental learning aims to acquire new categories over time while preserving prior knowledge, with catastrophic forgetting as the core challenge. Existing methods can be categorized into four main groups. First, output-level distillation (He et al., 2025b; Wang et al., 2025b; Rebuffi et al., 2017) transfers the old model's logits and features to the new one to curb prediction drift. Second, parameter regularization (Wang et al., 2025a; Jung et al., 2020), exemplified by elastic weight consolidation (Kirkpatrick et al., 2017), penalizes changes to important weights so new learning does not overwrite old knowledge. Third, replay or exemplar memory (Aljundi et al., 2019a;b; Zhou et al., 2024b) stores a small set of representative samples or uses generative replay to stabilize the decision boundary. Fourth, parameter isolation and structural expansion (He et al., 2026; 2025a; Yan et al., 2021; Li et al., 2019b) allocate task-specific capacity through masks, sub-networks, or expandable branches to reduce interference between old and new knowledge.

### 2.2. Incremental Object Detection

Incremental Object Detection (IOD) adapts detectors to new categories while retaining previously learned knowledge. Unlike continual learning for classification, where each task uses a fixed label set, IOD operates on images that contain both old and new objects while only the new categories are annotated. This annotation mismatch causes unlabeled old-class instances to be suppressed as background or misassigned to new classes.

Most incremental object detection methods follow a consistent paradigm across different detector architectures. Single-stage detectors like GFL (Li & Hoiem, 2017; Li et al., 2019a; Peng et al., 2021; Feng et al., 2022; Wang et al., 2025c; Zhang et al., 2024), two-stage frameworks such as Faster R-CNN (Liu et al., 2023a; Mo et al., 2024), and transformer-based methods (Liu et al., 2023b; Kang et al., 2023; Zhang et al., 2024), all employ pseudo-labeling from previous models to identify old-class instances while filtering out regions potentially containing new categories. In transformer-based methods, CL-DETR (Liu et al., 2023b) selects reliable pseudo labels through dual filtering on IoU and confidence, and BPF (Mo et al., 2024) adopts a similar strategy with multiple teachers on Faster R-CNN (Ren et al., 2016). Subsequent work enhances this foundation through synthetic exemplar generation using Stable Diffusion (Kim et al., 2024) and improved pseudo-label filtering

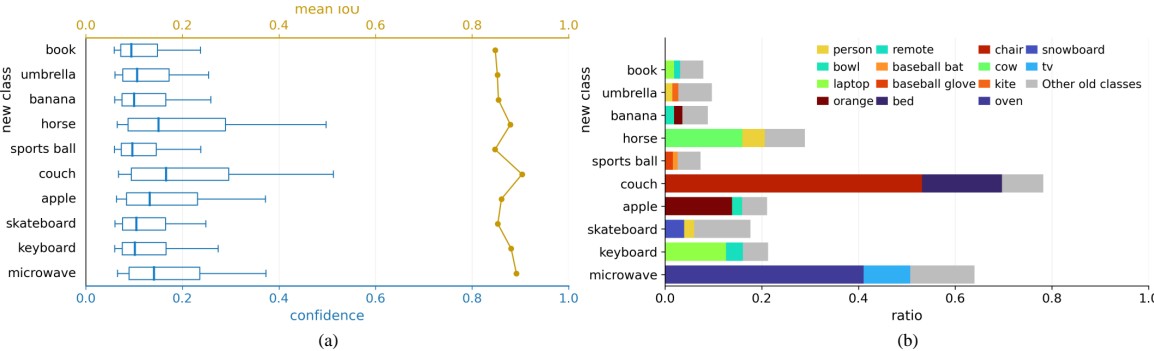

*Figure 2.* Statistical analysis on COCO 2017 under the 70+10 setting, using old-detector predictions with $\text{IoU} > 0.7$ to new-class ground truth. (a) Confidence distribution and mean IoU of the old detector's old-class predictions on new-class ground-truth instances. (b) Per-class proportion of new-class ground-truth instances misclassified as old classes by the old detector.

techniques (Wang et al., 2025c). DCA (Zhang et al., 2025a) introduces a localization-then-recognition paradigm that decouples localization from recognition to reduce forgetting, and GCD (Wang et al., 2025d) incorporates language priors through textual grounding. However, these methods share a fundamental limitation: they treat high-confidence, low-IoU detections as exclusively clean old-class evidence, thereby overlooking the inherent object symbiosis in detection scenarios. In contrast, our SIKD framework explicitly embraces object symbiosis, maintaining a unified feature space and modeling inter-object dependencies rather than suppressing them.

## 3. Methodology

### 3.1. Problem Formulation

In incremental object detection, the detector is trained over $T$ tasks. The class domain is $\mathcal{C} = \bigcup_{i=1}^{T} \mathcal{C}^i$ with $\mathcal{C}^i \cap \mathcal{C}^j = \emptyset$ for different tasks $i$ and $j$. The dataset is $\mathcal{D} = \bigcup_{i=1}^{T} \mathcal{D}^i$, where each $\mathcal{D}^i$ provides annotations $\mathcal{Y}^i$ only for classes in $\mathcal{C}^i$. At task $t$, the model $\mathcal{M}^{t-1}$ is updated to $\mathcal{M}^t$ using only $\mathcal{D}^t$ and $\mathcal{Y}^t$. Images in $\mathcal{D}^t$ may still contain unlabeled instances from previously learned classes $\mathcal{C}^{1:t-1} = \bigcup_{i=1}^{t-1} \mathcal{C}^i$. The objective is to learn the new classes $\mathcal{C}^t$ while maintaining performance on $\mathcal{C}^{1:t}$ without accessing earlier data $\{\mathcal{D}^1, \ldots, \mathcal{D}^{t-1}\}$.

### 3.2. Transformer-based Detectors

Following CL-DETR (Liu et al., 2023b), we adopt Deformable DETR (Zhu et al., 2020) as the architecture. In Deformable DETR, a transformer encoder processes image features, and the decoder operates on a set of $n$ learnable object queries $\mathcal{Q} = [\mathbf{q}_1, \ldots, \mathbf{q}_n]^\top \in \mathbb{R}^{n \times d}$. Each query hypothesizes one object and gathers evidence from the encoded features via cross-attention. A prediction head maps the decoded queries to class logits $\mathbf{z}_i \in \mathbb{R}^e$ and a class-agnostic box $\mathbf{b}_i \in \mathbb{R}^4$, where $e$ is the number of categories.

The decoder of DETR has $L$ layers. At each layer, queries are refined by self-attention, multi-scale deformable cross-attention, and a feed-forward block. Learned reference points are updated across layers, enabling progressive localization and classification refinement. Intermediate predictions are produced at every layer, and the final outputs after $L$ layers are the refined query embeddings $\mathcal{Q}^{(L)}$ together with logits $\{\mathbf{z}_i\}_{i=1}^n$ and boxes $\{\mathbf{b}_i\}_{i=1}^n$.

### 3.3. Symbiosis-aware Query Partitioning

As shown in Fig. 2, incremental object detection naturally exhibits object symbiosis, where old and new categories co-occur and occlude each other in the current training data. When the old model $\mathcal{M}^{t-1}$ processes current data $\mathcal{D}^t$, these relationships emerge as structured patterns in the query space. New objects often activate queries of semantically similar old classes, while partially visible old objects still trigger relevant query responses based on visible cues and context.

At step $t$, we use $\mathcal{M}^{t-1}$ predictions on $\mathcal{D}^t$ to guide the new model $\mathcal{M}^t$. Let the old model's queries be $\mathcal{Q}^{t-1} = [\mathbf{q}_1^{t-1}, \ldots, \mathbf{q}_n^{t-1}]^\top$ with outputs $\mathbf{z}_i^{t-1}$ for logits and $\mathbf{b}_i^{t-1}$ for the box of each query $\mathbf{q}_i^{t-1}$. The ground truth for $\mathcal{D}^t$ is

$$\mathcal{Y}^t = \{(\mathbf{g}_j^t, o_j^t)\}_{j=1}^{|\mathcal{Y}^t|}, \tag{1}$$

where $\mathbf{g}_j^t$ denotes a bounding box with label $o_j^t$.

Let $\sigma(\cdot)$ denote the sigmoid. For each query we define old class confidence and the maximum overlap as

$$\begin{aligned} s_i^{t-1} &= \max \sigma(\mathbf{z}_i^{t-1}), \\ v_i^{t-1} &= \max_{\forall (\mathbf{g}_j^t, o_j^t) \in \mathcal{Y}^t} \text{IoU}(\mathbf{b}_i^{t-1}, \mathbf{g}_j^t). \end{aligned} \tag{2}$$

With confidence threshold $\gamma$ and IoU threshold $\tau$, we parti-

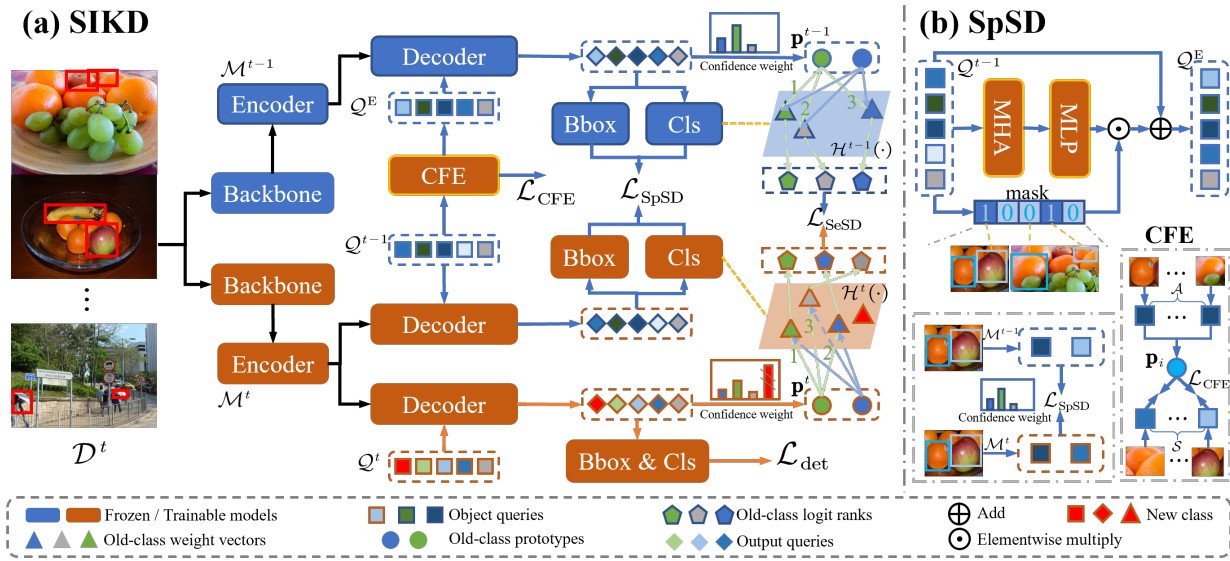

*Figure 3.* Overview of our proposed SIKD. (a) Training pipeline. The frozen old model $\mathcal{M}^{t-1}$ produces queries $\mathcal{Q}^{t-1}$ on $\mathcal{D}^t$. CFE refines queries of symbiotic regions under anchor-prototype guidance, yielding $\mathcal{Q}^{\mathrm{E}}$ that reduces old-class bias and removes redundancy. SpSD distills anchor logits and boxes, which enforces confidence-weighted, layer-wise logit consistency over all queries. SeSD builds confidence-weighted, $L_2$-normalized old-class prototypes from the last decoder layer of both models and aligns their classifier ranks to preserve the topology of old classes. (b) SpSD module. CFE contains multi-head self-attention (MHA) and an MLP, which is optimized with a prototype-guided cosine loss $\mathcal{L}_{\mathrm{CFE}}$ on $\mathcal{Q}^{\mathrm{E}}$, and is discarded at inference.

tion the queries into index sets $\mathcal{A}, \mathcal{S}, \mathcal{R} \subseteq \{1, \ldots, n\}$:

$$
\begin{aligned}
\mathcal{A} &= \{\, i \mid s_i^{t-1} \geq \gamma \ \wedge \ v_i^{t-1} < \tau \,\}, \\
\mathcal{S} &= \{\, i \mid v_i^{t-1} \geq \tau \,\}, \\
\mathcal{R} &= \{\, i \mid s_i^{t-1} < \gamma \ \wedge \ v_i^{t-1} < \tau \,\}.
\end{aligned}
\tag{3}
$$

This partition makes object symbiosis explicit at the query level where DETR performs inference. The set $\mathcal{A}$ contains high-confidence, low-overlap detections of old classes, which serve as stable anchors. The set $\mathcal{S}$ collects queries from overlapping scenarios caused by co-occurrence and occlusion. It includes cases where unseen new objects are misclassified as semantically similar old classes and cases where partially visible old objects remain detectable. These queries form symbiotic regions that encapsulate spatial and semantic dependencies. In contrast to prior methods that often discard high overlap predictions, we retain and exploit them as valuable supervisory signals. The set $\mathcal{R}$ comprises low-confidence queries that still carry weak relational cues and are utilized with reduced weighting.

### 3.4. Overall Framework

We propose SIKD for incremental object detection with DETR-style detectors. At step $t$, the frozen model $\mathcal{M}^{t-1}$ runs on $\mathcal{D}^t$ to produce queries $\mathcal{Q}^{t-1}$ and predictions. Following Sec. 3.3, we partition the queries into stable anchors $\mathcal{A}$, symbiotic regions $\mathcal{S}$ and residual queries $\mathcal{R}$. The core of SIKD consist of two complementary distillation pathway.

Spatial Symbiosis Distillation refines $\mathcal{S}$ with CFE while keeping $\mathcal{A}$ fixed and yields the enhanced set $\mathcal{Q}^{\mathrm{E}}$. SpSD then applies slot aligned, confidence weighted supervision by feeding $\mathcal{Q}^{\mathrm{E}}$ to the frozen old decoder and $\mathcal{Q}^{t-1}$ to the new decoder. This promotes instance-level consistency in space. Semantic Symbiosis Distillation aggregates the last layer decoder outputs of both models into confidence weighted, $L_2$ normalized prototypes for each old class. It evaluates these prototypes in the old-class logit subspace and aligns their soft ranks to preserve semantic structure during adaptation. Together, SpSD and SeSD convert overlap-driven signals into reliable supervision and maintain a unified feature space across incremental tasks.

### 3.5. Spatial Symbiosis Distillation

Spatial Symbiosis Distillation (SpSD) leverages object symbiosis in the spatial dimension by exploiting co-occurrence and occlusion patterns that manifest as distinctive signatures in the feature space. As identified in Sec. 3.3, these symbiotic patterns concentrate in the overlap-driven set $\mathcal{S}$, while the set $\mathcal{A}$ provides semantically reliable anchors of old-class knowledge. Directly transferring knowledge from all old model queries causes two issues: It inflates old class confidence on unseen categories, and it suppresses responses to partially occluded old class objects, which leads to misclassification or background assignment. SpSD preserves anchor integrity and transforms symbiotic queries into anchor aligned representations, which maintains feature con-

sistency across spatially related instances.

**Consistent Feature Enhancement.** We aim to enhance spatially consistent features in symbiotic regions while preventing anchor drift. This requires modeling contextual relationships among queries to enable ambiguous symbiotic slots to aggregate evidence from reliable anchors. We refine queries using multi-head self-attention followed by an MLP while preserving anchors:

$$
\begin{aligned}
\Delta \mathcal{Q}^{t-1} &= \text{MLP}\big(\text{MHA}(\mathcal{Q}^{t-1})\big), \\
\mathbf{q}_i^{\text{E}} &= \mathbf{q}_i^{t-1} + \mathbb{1}_{i \notin \mathcal{A}}\, \Delta \mathbf{q}_i^{t-1}.
\end{aligned}
\tag{4}
$$

Here $\Delta \mathcal{Q}^{t-1} = [\Delta \mathbf{q}_1^{t-1}, \ldots, \Delta \mathbf{q}_n^{t-1}]^\top$, $\Delta \mathbf{q}_i^{t-1}$ is the $i$-th row of $\Delta \mathcal{Q}^{t-1}$, and $\mathbb{1}_{i \notin \mathcal{A}} = 1$ if $i \notin \mathcal{A}$ and $0$ otherwise.

To guide this enhancement, we construct class-consistent prototypes from anchors. For each enhanced query $\mathbf{q}_i^{\text{E}}$ with predicted class $y_i^{t-1}$, the confidence-weighted prototype is

$$
\mathbf{p}_i = \text{norm}\left( \frac{\sum_{j \in \mathcal{P}_{\mathcal{A}}(i)} s_j^{t-1} \mathbf{q}_j^{\text{E}}}{\sum_{j \in \mathcal{P}_{\mathcal{A}}(i)} s_j^{t-1} + \varepsilon} \right) \in \mathbb{R}^d,
\tag{5}
$$

where $\mathcal{P}_{\mathcal{A}}(i) = \{ j \in \mathcal{A} \mid y_j^{t-1} = y_i^{t-1},\ j \neq i \}$. We then align enhanced symbiotic queries to their prototypes with a cosine objective:

$$
\mathcal{L}_{\text{CFE}} = \frac{1}{|\mathcal{S}|} \sum_{i \in \mathcal{S}} \left( 1 - \text{norm}(\mathbf{q}_i^{\text{E}})^\top \mathbf{p}_i \right).
\tag{6}
$$

**Spatially Aligned Distillation.** Let $\mathcal{Q}^{\text{E}} = [\mathbf{q}_1^{\text{E}}, \ldots, \mathbf{q}_n^{\text{E}}]^\top$ denote the enhanced query set. We feed $\mathcal{Q}^{\text{E}}$ to the decoder of the frozen old model $\mathcal{M}^{t-1}$ and $\mathcal{Q}^{t-1}$ to the decoder of the new model $\mathcal{M}^t$, producing logits $\mathbf{z}_i^{\text{E}}, \hat{\mathbf{z}}_i^t \in \mathbb{R}^m$ and boxes $\mathbf{b}_i^{\text{E}}, \hat{\mathbf{b}}_i^t \in \mathbb{R}^4$, where $m = |\mathcal{C}^{1:t-1}|$ denotes the number of old classes. For anchors we distill semantics and geometry:

$$
\begin{aligned}
\mathcal{L}_A = \frac{1}{|\mathcal{A}|} \sum_{i \in \mathcal{A}} \Big[ &\mathcal{L}_{\text{KL}}\big(\hat{\mathbf{z}}_i^t, \mathbf{z}_i^{\text{E}}\big) + \lambda_1 \mathcal{L}_{L1}\big(\hat{\mathbf{b}}_i^t, \mathbf{b}_i^{\text{E}}\big) \\
&+ \lambda_2 \mathcal{L}_{\text{GIoU}}\big(\hat{\mathbf{b}}_i^t, \mathbf{b}_i^{\text{E}}\big) \Big].
\end{aligned}
\tag{7}
$$

Here $\mathcal{L}_{\text{KL}}$ is the KL divergence on the old-class logit subspace, $\mathcal{L}_{L1}$ is the $\ell_1$ loss on box coordinates, and $\mathcal{L}_{\text{GIoU}}$ is the generalized IoU loss.

To maintain consistency across decoder layers while handling prediction uncertainty, we employ layer-specific confidence weights at layer $\ell$ as:

$$
w_i^{(\ell)} = \frac{s_i^{t-1,(\ell)}}{\sum_j s_j^{t-1,(\ell)} + \varepsilon},
\tag{8}
$$

where $s_i^{t-1,(\ell)}$ denotes the old model confidence for query $i$ at decoder layer $\ell$. The layer wise distillation objective

enforces progressive feature alignment:

$$
\mathcal{L}_{\text{ID}} = \sum_{\ell=1}^L \sum_i w_i^{(\ell)} \left\| \mathbf{z}_i^{\text{E},(\ell)} - \hat{\mathbf{z}}_i^{t,(\ell)} \right\|_2^2.
\tag{9}
$$

The overall spatial distillation objective combines both components:

$$
\mathcal{L}_{\text{SpSD}} = \mathcal{L}_A + \alpha \mathcal{L}_{\text{ID}},
\tag{10}
$$

where $\alpha$ balances the contributions from anchor distillation and layer-wise feature alignment. With this weighting, SpSD preserves spatial coherence by aligning symbiotic regions during incremental learning and reduces representation drift by turning object co-occurrence and occlusion into useful supervision.

### 3.6. Semantic Symbiosis Distillation

While SpSD preserves instance-level relationships, its effectiveness diminishes as query assignments shift during incremental training. Semantic Symbiosis Distillation (SeSD) addresses this limitation by transitioning to class-level structure preservation, maintaining the semantic topology of old classes through prototype-based alignment that remains robust to instance-level correspondence changes.

SeSD constructs stable class representations by aggregating features into confidence-weighted prototypes. For each old class $c \in \mathcal{C}^{1:t-1}$ across both model states:

$$
\mathbf{p}_c^\pi = \text{norm}\left( \frac{\sum_{j \in \mathcal{P}^\pi(c)} s_j^\pi \mathbf{q}_j^{\pi,(L)}}{\sum_{j \in \mathcal{P}^\pi(c)} s_j^\pi + \varepsilon} \right), \quad \pi \in \{t-1, t\},
\tag{11}
$$

where $\pi$ indexes the model state ($t-1$ for old model, $t$ for new model), $\mathcal{P}^\pi(c)$ denotes the set of queries assigned to old class $c$ by model $\mathcal{M}^\pi$, and $s_j^\pi$ represents the corresponding confidence score derived from old-class logits.

To construct semantic relations among old classes, we project prototypes through the classifier heads $\mathcal{H}^\pi(\cdot)$ ($\pi \in \{t-1, t\}$) and normalize the resulting score vectors:

$$
\tilde{\mathbf{s}}_c^\pi = \frac{\sigma\big(\mathcal{H}^\pi(\mathbf{p}_c^\pi)\big)_{1:m}}{\max \big[\sigma\big(\mathcal{H}^\pi(\mathbf{p}_c^\pi)\big)_{1:m}\big]} \in \mathbb{R}^m,
\tag{12}
$$

where $m = |\mathcal{C}^{1:t-1}|$ denotes the total number of old classes, and the new model is restricted to these old-class dimensions to prevent interference from new categories.

We align semantic structures by the distillation objective:

$$
\mathcal{L}_{\text{SeSD}} = \frac{1}{(m)^2} \sum_{c=1}^m \left\| \text{rank}(\tilde{\mathbf{s}}_c^t) - \text{rank}(\tilde{\mathbf{s}}_c^{t-1}) \right\|_1,
\tag{13}
$$

where $\text{rank}(\tilde{\mathbf{s}}_c^t)_k = \sum_{j=1}^m \sigma(-(\tilde{s}_{c,k}^t - \tilde{s}_{c,j}^t))$ computes the soft rank position of class $k$ within the score vector, representing its relative semantic ordering. This rank-based

*Table 1.* Experimental results (%) on the COCO 2017 two-task settings. Best results are in **bold**. Methods marked with * use exemplars.

| Setting | Method | Baseline | $AP$ | $AP_{50}$ | $AP_{75}$ | $AP_S$ | $AP_M$ | $AP_L$ |
|---------|--------|----------|------|-----------|-----------|--------|--------|--------|
| 70 + 10 | LwF (Li & Hoiem, 2017) | GFLv1 | 7.1 | 12.4 | 7.0 | 4.8 | 9.5 | 10.0 |
|  | RILOD (Li et al., 2019a) | GFLv1 | 24.5 | 37.9 | 25.7 | 14.2 | 27.4 | 33.5 |
|  | SID (Peng et al., 2021) | GFLv1 | 32.8 | 49.0 | 35.0 | 17.1 | 36.9 | 44.5 |
|  | ERD (Feng et al., 2022) | GFLv1 | 34.9 | 51.9 | 37.4 | 18.7 | 38.8 | 45.5 |
|  | TLR (Zhang et al., 2024) | GLIP | 42.9 | 59.2 | 45.2 | 24.3 | 45.1 | 54.1 |
|  | CL-DETR* (Liu et al., 2023b) | Deformable DETR | 40.4 | 58.0 | 43.9 | 23.8 | 43.6 | 53.5 |
|  | DyQ-DETR* (Zhang et al., 2025b) | Deformable DETR | 42.4 | 60.4 | 46.3 | 24.5 | 45.7 | 57.5 |
|  | CL-DETR (Liu et al., 2023b) | Deformable DETR | 35.8 | 53.5 | 39.5 | 19.4 | 41.5 | 46.1 |
|  | ACF (Kang et al., 2023) | Deformable DETR | 37.6 | – | – | – | – | – |
|  | DCA (Zhang et al., 2025a) | Deformable DETR | 41.3 | 59.2 | – | – | – | – |
|  | DyQ-DETR (Zhang et al., 2025b) | Deformable DETR | 39.5 | 56.4 | 43.1 | 22.5 | 43.1 | 53.0 |
|  | SIKD (Ours) | Deformable DETR | **44.3** | **62.9** | **47.8** | **28.2** | **47.7** | **59.5** |
| 40 + 40 | LwF (Li & Hoiem, 2017) | GFLv1 | 17.2 | 25.4 | 18.6 | 7.9 | 18.4 | 24.3 |
|  | RILOD (Li et al., 2019a) | GFLv1 | 29.9 | 45.0 | 32.0 | 15.8 | 33.0 | 40.5 |
|  | SID (Peng et al., 2021) | GFLv1 | 34.0 | 51.4 | 36.3 | 18.4 | 38.4 | 44.9 |
|  | ERD (Feng et al., 2022) | GFLv1 | 36.9 | 54.5 | 39.6 | 21.3 | 40.4 | 47.5 |
|  | TLR (Zhang et al., 2024) | GLIP | 40.4 | 57.4 | 43.9 | 23.3 | 44.7 | 54.5 |
|  | CL-DETR* (Liu et al., 2023b) | Deformable DETR | 42.0 | 60.1 | 45.9 | 24.0 | 45.3 | 55.6 |
|  | DyQ-DETR* (Zhang et al., 2025b) | Deformable DETR | 42.4 | 60.5 | 45.9 | 23.9 | 46.3 | 56.7 |
|  | CL-DETR (Liu et al., 2023b) | Deformable DETR | 39.2 | 56.1 | 42.6 | 21.0 | 42.8 | 52.6 |
|  | ACF (Kang et al., 2023) | Deformable DETR | 39.8 | – | – | – | – | – |
|  | DCA (Zhang et al., 2025a) | Deformable DETR | 42.8 | 58.4 | – | – | – | – |
|  | DyQ-DETR (Zhang et al., 2025b) | Deformable DETR | 41.4 | 59.7 | 44.9 | 24.1 | 45.2 | 54.3 |
|  | SIKD (Ours) | Deformable DETR | **43.3** | **61.7** | **46.7** | **26.4** | **46.6** | **57.0** |

alignment preserves topological relationships independent of absolute confidence values (Tao et al., 2020; Liu et al., 2022), focusing on the essential semantic structure rather than magnitude variations.

This semantic alignment preserves the relative logit structure of old classes, anchoring $\mathcal{Q}^{t,(L)}$ to $\mathcal{Q}^{t-1,(L)}$ even when instance correspondences break down. By complementing spatial distillation with semantic structure preservation, SeSD stabilizes the feature space throughout tasks.

### 3.7. Training Objective

The training objective combines our symbiotic distillation terms with the standard detection loss. This joint objective maintains performance across incremental steps:

$$\mathcal{L}_{\text{total}} = \underbrace{\mathcal{L}_{\text{det}} + \mathcal{L}_{\text{SpSD}} + \beta\mathcal{L}_{\text{SeSD}}}_{\mathcal{L}_{\text{model}}} + \mathcal{L}_{\text{CFE}}. \qquad (14)$$

Here $\mathcal{L}_{\text{det}}$ is the standard DETR detection loss, $\mathcal{L}_{\text{SpSD}}$ maintains spatial consistency through query-level alignment, $\mathcal{L}_{\text{SeSD}}$ preserves semantic structure via prototype-based ranking, and $\mathcal{L}_{\text{CFE}}$ enhances features in symbiotic regions. The model parameters are updated using $\mathcal{L}_{\text{model}}$, while only the CFE module is optimized via $\mathcal{L}_{\text{CFE}}$, with

gradients isolated between these components.

## 4. Experiments

### 4.1. Experimental Settings

**Datasets and Evaluation Metrics.** Consistent with prior work (Liu et al., 2023b; Kim et al., 2024; Zhang et al., 2025b;a), we adopt the standard COCO 2017 (Lin et al., 2014) evaluation protocol and incremental-setting notation. We evaluate on COCO 2017, which contains 80 object categories with 118k training images and 5k validation images. Performance follows the standard COCO metrics. The primary metric, $AP$, is the average precision averaged over IoU thresholds from 0.50 to 0.95 in steps of 0.05. We also report $AP_{50}$ and $AP_{75}$ at single IoU thresholds of 0.50 and 0.75. Scale-specific scores $AP_S$, $AP_M$, and $AP_L$ evaluate small, medium, and large objects, where small means area $< 32^2$ pixels, medium means $32^2 \leq$ area $< 96^2$, and large means area $\geq 96^2$ pixels. For incremental settings denoted A + B, we follow prior work: the initial step contains A classes, and each subsequent step adds B new classes.

**Implementation Details.** We implement our method within MMDetection on Deformable DETR with a ResNet-50 back-

*Table 2.* Experimental results ($AP$ / $AP_{50}$, %) on the COCO 2017 multi-task settings. Methods marked with * use exemplars.

| Method | $(1-40)$ | 40+10+10+10+10 | | | | 40+20+20 | |
| --- | --- | --- | --- | --- | --- | --- | --- |
| | | + $(40-50)$ | + $(50-60)$ | + $(60-70)$ | + $(70-80)$ | + $(40-60)$ | + $(60-80)$ |
| RILOD (Li et al., 2019a) | 45.7 / 66.3 | 25.4 / 38.9 | 11.2 / 17.3 | 10.5 / 15.6 | 8.4 / 12.5 | 27.8 / 42.8 | 15.8 / 4.0 |
| SID (Peng et al., 2021) | 45.7 / 66.3 | 34.6 / 52.1 | 24.1 / 38.0 | 14.6 / 23.0 | 12.6 / 23.3 | 34.0 / 51.8 | 23.8 / 36.5 |
| ERD (Feng et al., 2022) | 45.7 / 66.3 | 36.4 / 53.9 | 30.8 / 46.7 | 26.2 / 39.9 | 20.7 / 31.8 | 36.7 / 54.6 | 32.4 / 48.6 |
| CL-DETR* (Liu et al., 2023b) | 46.5 / 68.6 | – / – | – / – | – / – | 28.1 / – | – / – | 35.3 / – |
| ACF (Kang et al., 2023) | 48.0 / – | 39.1 / – | 35.4 / – | 32.0 / – | 30.3 / – | 39.3 / – | 36.6 / – |
| DCA (Zhang et al., 2025a) | 48.0 / 68.9 | **44.0** / 61.2 | 41.1 / 56.5 | 39.2 / 53.8 | 37.2 / 49.6 | 42.7 / 59.6 | 40.3 / 54.1 |
| SIKD (Ours) | 45.4 / 64.7 | 43.6 / **62.3** | **41.1 / 59.9** | **39.8 / 57.8** | **38.1 / 55.5** | **43.4 / 62.1** | **40.8 / 58.8** |

*Table 3.* Ablations on COCO 2017 (70+10) with Deformable DETR. "All categories" reports the $AP$ of the final-phase model over all 80 categories. "Old categories" reports the $AP$ of the final-phase model on the 70 categories introduced in phase 1. "FPP" is the $AP$ difference between the phase-1 model and the final-phase model on those 70 categories (lower is better). Idx 5 corresponds to our method.

| Idx | Raw KD | SpSD | SeSD | All categories ↑ | | | Old categories ↑ | | | FPP ↓ | | |
| --- | --- | --- | --- | --- | --- | --- | --- | --- | --- | --- | --- | --- |
| | | | | $AP$ | $AP_{50}$ | $AP_{75}$ | $AP$ | $AP_{50}$ | $AP_{75}$ | $AP$ | $AP_{50}$ | $AP_{75}$ |
| 1 | | | | 41.2 | 59.6 | 44.4 | 41.5 | 60.1 | 44.7 | 4.9 | 5.4 | 5.4 |
| 2 | ✓ | | | 41.1 | 59.4 | 44.3 | 41.3 | 59.9 | 44.5 | 5.1 | 5.6 | 5.6 |
| 3 | | ✓ | | 42.3 | 60.8 | 45.8 | 42.8 | 61.6 | 46.2 | 3.6 | 3.9 | 3.9 |
| 4 | | | ✓ | 43.9 | 62.6 | 47.4 | 44.4 | 63.4 | 47.9 | 2.0 | 2.1 | 2.2 |
| 5 | | ✓ | ✓ | 44.3 | 62.9 | 47.8 | 45.1 | 64.0 | 48.7 | 1.3 | 1.5 | 1.4 |

bone pre-trained on ImageNet. All experiments run on four RTX 4090 GPUs, and the basic training settings follow the official implementation (Chen et al., 2019). We use fixed hyperparameters across all settings. Following prior work (Wang et al., 2024), we set $\lambda_1 = 5.0$ and $\lambda_2 = 2.0$. We set $\alpha = 1.0$ and $\beta = 6.0$ as our choices. To ensure comparability and reproducibility, we randomize the category order using the predefined random seed released with CL-DETR (Liu et al., 2023b) and adopt the resulting order. The pseudo-label selection threshold in each incremental phase is $0.4$, and the IoU threshold is $0.7$.

### 4.2. Comparison with the State-of-the-Arts

**Two-task settings.** Table 1 compares our method with prior state-of-the-art methods on the COCO 2017 two-task splits. Compared with DyQ-DETR and DCA, both implemented on Deformable DETR, our method improves $AP$ by 1.9 and 3.0 points on the 70+10 split and by 0.9 and 0.5 points on the 40+40 split. The corresponding $AP_{50}$ gains are 2.5 and 3.7 points on 70+10 and 1.2 and 3.3 points on 40+40. It also surpasses the GLIP-based TLR, with gains of 1.5 and 2.9 $AP$ points and 3.7 and 4.3 $AP_{50}$ points on 70+10 and 40+40, respectively.

**Multi-task settings.** Table 2 reports results on COCO 2017 under the multi-task settings. In the initial base training phase, our implementation achieves slightly lower performance than some prior methods. However, our method establishes new state-of-the-art results in all subse-

quent incremental phases. Compared with DCA, under the 40+10+10+10+10 setting we improve $AP$ by 0.9 and $AP_{50}$ by 5.9 points. Under the 40+20+20 setting we improve $AP$ by 0.5 and $AP_{50}$ by 4.7. The improvements reflect superior retention of prior knowledge coupled with efficient acquisition of new concepts.

### 4.3. Results and Analysis

**Results on DIOR dataset.** We further evaluate our method on the DIOR dataset (Li et al., 2020). DIOR is a large-scale optical remote sensing benchmark with 20 categories and strong variation in scale, viewpoint, object density, and background, where co-occurrence and occlusion are common. We follow three two-task settings 10+10, 15+5, and 19+1 and report $AP_{50}$ as summarized in Table 4. Compared with the replay-based method CL-DETR*, SIKD improves the All score by 6.2 points in the 10+10 setting, by 7.3 points in the 15+5 setting, and by 9.0 points in the 19+1 setting.

**Ablation study of SIKD.** We evaluate the contribution of each component under the 70+10 setting, with results summarized in Table 3. Idx 1 is the baseline, which adopts a standard pseudo-labeling strategy without any of our proposed modules. Idx 2 (Raw KD) incorporates high-IoU queries directly into the distillation process. Idx 3 employs only SpSD, and Idx 4 uses only SeSD. Idx 5 combines both SpSD and SeSD, representing our full SIKD method.

Compared to the baseline in Idx 1, directly integrating high-

*Table 4.* Experimental results (%) on the DIOR dataset under the two-task settings. $AP_{50}$ is reported for Old, New, All and task-averaged (Avg). Best results are in **bold**. Methods marked with * use exemplars.

| Setting | Methods | Old | New | All | Avg |
|---------|---------|-----|-----|-----|-----|
| 10+10 | CL-DETR | 42.2 | 63.9 | 53.1 | 53.1 |
| | CL-DETR* | 64.4 | 61.5 | 63.0 | 63.0 |
| | SIKD (Ours) | **72.0** | **66.4** | **69.2** | **69.2** |
| 15+5 | CL-DETR | 43.0 | 63.8 | 48.2 | 53.4 |
| | CL-DETR* | 64.3 | 60.4 | 63.4 | 62.4 |
| | SIKD (Ours) | **71.8** | **67.5** | **70.7** | **69.6** |
| 19+1 | CL-DETR | 47.6 | 44.0 | 47.5 | 45.8 |
| | CL-DETR* | 57.3 | 53.0 | 57.1 | 55.2 |
| | SIKD (Ours) | **65.3** | **79.7** | **66.1** | **72.5** |

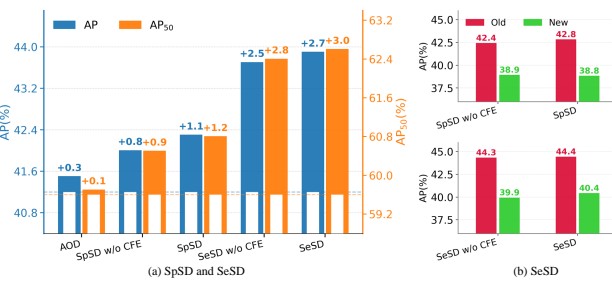

*Figure 4.* Ablations of SpSD and SeSD on COCO 2017 (70+10). In (a), the inner bars denote the baseline, with the full bars showing absolute values and labels indicating improvements over baseline.

overlap queries in Index 2 causes a drop of 0.1 $AP$, suggesting that such queries introduce noise and bias when used naively. Using SpSD alone (Idx 3) improves $AP$ by 1.1 points, while SeSD alone (Idx 4) brings a more substantial gain of 2.7 points, highlighting the individual efficacy of each distillation pathway. The full SIKD model (Idx 5) achieves the best performance of $44.3$ $AP$, demonstrating that combining spatial and semantic distillation yields complementary benefits and the highest overall accuracy.

**Analysis of SpSD and SeSD.** In Fig. 4(a), Anchor-Only Distillation (AOD) using Eq. 7 raises $AP$ by 0.3, indicating a modest gain. SpSD without CFE improves $AP$ by 0.8, showing that overlap-driven evidence is useful. Adding CFE to SpSD lifts the gain to 1.1, which further stabilizes instance-level consistency. SeSD without CFE improves $AP$ by 2.5, highlighting the value of preserving class-level topology. SeSD with CFE achieves 2.7, confirming that feature enhancement also benefits the class-level objective. In Fig. 4(b) and (c), CFE contributes 0.4 $AP$ to SpSD on the old task and 0.5 $AP$ to SeSD on the new task.

**Analysis of the balance weight.** As shown in Table 5, we ablate the balancing coefficients $\beta$ and $\alpha$ on COCO 2017 (70+10). For $\beta$, increasing the weight from 4 to 8 raises All

*Table 5.* Ablation of the balance weights on COCO 2017 (70+10). $AP$ is reported for Old, New, All and task-averaged (Avg).

| Setting | Old | New | All | Avg |
|---------|-----|-----|-----|-----|
| $\beta = 4$ | 44.7 | 39.1 | 44.0 | 41.9 |
| $\beta = 6$ | 45.1 | 38.8 | 44.3 | 42.0 |
| $\beta = 8$ | 45.2 | 38.5 | 44.4 | 41.9 |
| $\alpha = 0.1$ | 42.1 | 39.9 | 41.9 | 41.0 |
| $\alpha = 1$ | 42.8 | 38.8 | 42.3 | 40.8 |
| $\alpha = 10$ | 43.2 | 34.9 | 42.2 | 39.1 |

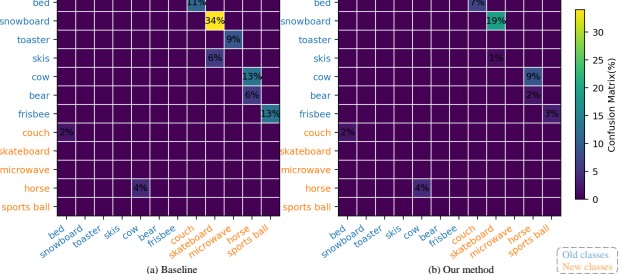

*Figure 5.* Comparison of confusion matrices for old and new Classes between the baseline and our method on COCO (70+10).

$AP$ by 0.4 and Old $AP$ by 0.5, while reducing New $AP$ by 0.6. Setting $\beta = 6$ yields the best task-averaged $AP$ (42.0) and provides a favorable trade-off between retention and plasticity, so we adopt $\beta = 6$ in the main experiments. For $\alpha$, larger values bias training toward preserving old knowledge. Old $AP$ increases by 1.1 as $\alpha$ grows from 0.1 to 10, whereas New $AP$ drops by 5.0 and Avg $AP$ decreases by 1.9. Setting $\alpha = 1$ achieves the highest All $AP$ (42.3) with a reasonable balance between old and new, and is therefore our default.

**Visualizations.** As shown in Fig. 5, we compare the confusion matrices between old and new categories for SIKD and the baseline on COCO 2017 (70+10). Relative to the baseline, SIKD reduces old-to-new confusion while not increasing new-to-old errors. These results support our design of maintaining a unified feature space across old and new classes, thereby mitigating the tendency of old-class knowledge to overfit toward new classes. The appendix includes additional experiments and visual analyses. Table 9 reports efficiency results, Table 6 provides further analysis of SeSD, Table 7 ablates the hyperparameters $\gamma$ and $\tau$, and Table 8 reports results under the multi-task setting. The appendix also contains Fig. 7 and additional qualitative visualizations.

## 5. Conclusion

We revisit IOD through object symbiosis and show that a unified feature space reduces confusion between old and new classes and curbs forgetting. SIKD models spatial sym-

biosis and semantic symbiosis. SpSD captures spatial symbiosis by refining symbiotic queries under anchor guidance and enforcing slot-aligned instance consistency. SeSD preserves semantic symbiosis by building confidence-weighted prototypes and aligning their ranks within the old-class subspace. Together, they convert overlap responses into reliable supervision and stabilize both spatial and semantic representations. Extensive experiments show consistent gains over state-of-the-art methods. In future work, we will explore using LLMs (Cheng et al., 2026; Xu et al., 2026) to inject contextual priors to further strengthen symbiosis-aware distillation.

## Acknowledgments

This work was supported in part by the National Key R&D Program of China under Grant No.2023YFA1008600, in part by the National Natural Science Foundation of China under Grants 62576262, U22A2096, in part by the Key Research and Development Program of Shaanxi Province under grant 2024SF-YBXM-647, in part by the Fundamental Research Funds for the Central Universities under Grant QTZX25083, QTZX23042.

## Impact Statement

This paper presents work whose goal is to advance the field of Machine Learning. There are many potential societal consequences of our work, none of which we feel must be specifically highlighted here.

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

## A. Training pipeline for SIKD

Algorithm 1 illustrates the training details of our method.

---

**Algorithm 1** SIKD training in new task $t$

---

**Require:** Frozen old model $\mathcal{M}^{t-1}$, new model $\mathcal{M}^t$, current dataset $\mathcal{D}^t$, thresholds $\gamma, \tau$, weights $\alpha, \beta$.

1: **for** each mini-batch $(X, \mathcal{Y}^t)$ in $\mathcal{D}^t$ **do**
2:     **Old-model forward and statistics**
3:     $\mathcal{Q}^{t-1}, \{\mathbf{z}_i^{t-1}\}_{i=1}^n$, and $\{\mathbf{b}_i^{t-1}\}_{i=1}^n \leftarrow \mathcal{M}^{t-1}(X)$
4:     **for** $i = 1$ to $n$ **do**
5:         $s_i^{t-1} \leftarrow \max \sigma(\mathbf{z}_i^{t-1})$, and $v_i^{t-1} \leftarrow \max_j \text{IoU}(\mathbf{b}_i^{t-1}, \mathbf{g}_j^t)$
6:     **end for**
7:     $\mathcal{A} \leftarrow \{\, i \mid s_i^{t-1} \geq \gamma \wedge v_i^{t-1} < \tau \,\}, \mathcal{S} \leftarrow \{\, i \mid v_i^{t-1} \geq \tau \,\}$, and $\mathcal{R} \leftarrow \{\, i \mid s_i^{t-1} < \gamma \wedge v_i^{t-1} < \tau \,\}$
8:     **Consistent Feature Enhancement (CFE)**
9:     $\Delta \mathcal{Q}^{t-1} \leftarrow \text{MLP}(\text{MHA}(\mathcal{Q}^{t-1}))$
10:     **for** $i = 1$ to $n$ **do**
11:         **if** $i \in \mathcal{A}$ **then**
12:             $\mathbf{q}_i^{\text{E}} \leftarrow \mathbf{q}_i^{t-1}$
13:         **else**
14:             $\mathbf{q}_i^{\text{E}} \leftarrow \mathbf{q}_i^{t-1} + \Delta \mathbf{q}_i^{t-1}$
15:         **end if**
16:     **end for**
17:     $\mathcal{Q}^{\text{E}} \leftarrow [\mathbf{q}_1^{\text{E}}, \ldots, \mathbf{q}_n^{\text{E}}]^\top$
18:     Build anchor-based prototypes $\{\mathbf{p}_i\}$ and compute $\mathcal{L}_{\text{CFE}}$ as in Eq. (6)
19:     **Spatial Symbiosis Distillation (SpSD)**
20:     $\{\mathbf{z}_i^{\text{E}}, \mathbf{b}_i^{\text{E}}\}_{i=1}^n \leftarrow$ decoder of $\mathcal{M}^{t-1}(\mathcal{Q}^{\text{E}})$, and $\{\hat{\mathbf{z}}_i^t, \hat{\mathbf{b}}_i^t\}_{i=1}^n \leftarrow$ decoder of $\mathcal{M}^t(\mathcal{Q}^{t-1})$
21:     Compute anchor loss $\mathcal{L}_A$ using Eq. (7)
22:     Compute layer-wise loss $\mathcal{L}_{\text{ID}}$ with $w_i^{(\ell)}$ from Eq. (9)
23:     $\mathcal{L}_{\text{SpSD}} \leftarrow \mathcal{L}_A + \alpha \mathcal{L}_{\text{ID}}$
24:     **Semantic Symbiosis Distillation (SeSD)**
25:     Build confidence-weighted prototypes $\{\mathbf{p}_c^{t-1}, \mathbf{p}_c^t\}$ for each old class $c$
26:     Obtain $\tilde{\mathbf{s}}_c^{t-1}, \tilde{\mathbf{s}}_c^t$ and compute $\mathcal{L}_{\text{SeSD}}$ via Eq. (13)
27:     **Total losses**
28:     $\hat{\mathcal{Y}}^t \leftarrow \mathcal{M}^t(X)$
29:     $\mathcal{L}_{\text{det}} \leftarrow \mathcal{L}_{\text{det}}(\hat{\mathcal{Y}}^t, \mathcal{Y}^t)$, and $\mathcal{L}_{\text{model}} \leftarrow \mathcal{L}_{\text{det}} + \mathcal{L}_{\text{SpSD}} + \beta \mathcal{L}_{\text{SeSD}}$
30:     Update the new detector with $\mathcal{L}_{\text{model}}$ and the CFE module with $\mathcal{L}_{\text{CFE}}$, while blocking gradients from each loss to the other branch.
31: **end for**
32: **Output:** updated new model $\mathcal{M}^t$

---

## B. Effect of rank alignment in SeSD

Table 6 compares rank alignment with direct score distillation on COCO 2017 (70+10). Using rank alignment improves all metrics. Old rises from 42.4 to 44.4 (+2.0), New from 39.6 to 40.4 (+0.8), All from 42.1 to 43.9 (+1.8), and Avg from 41.0 to 42.4 (+1.4). These gains indicate that preserving the relative ordering of class scores is more robust than matching raw scores. Rank alignment reduces sensitivity to calibration and scale differences between models, which helps maintain the semantic topology of old classes while adapting to new ones.

## C. Sensitivity to $\gamma$ and $\tau$

We analyze the sensitivity of SIKD to the two thresholds $\gamma$ and $\tau$ used in our symbiosis-aware query partitioning. As shown in Table 7, the performance is stable across a reasonable range of values. For $\gamma$, setting it to $0.4$ yields the best overall performance, while values $0.3$ and $0.5$ lead to only minor changes. For $\tau$, the default choice $\tau = 0.7$ achieves the highest

*Table 6.* SeSD ablation on COCO 2017 (70+10) comparing rank alignment with direct score distillation. *AP* (%) is reported for Old, New, All, and task averaged (Avg).

| Methods | Old | New | All | Avg |
|---|---|---|---|---|
| SeSD w/o rank | 42.4 | 39.6 | 42.1 | 41.0 |
| SeSD | 44.4 | 40.4 | 43.9 | 42.4 |

*Table 7.* Sensitivity analysis of thresholds $\gamma$ and $\tau$ on DIOR under the 15+5 incremental setting. $AP_{50}$ (%) is reported for old classes (Old), new classes (New), and all classes (All).

| Setting | Old | New | All |
|---|---|---|---|
| $\gamma = 0.3$ | 70.9 | 66.9 | 69.9 |
| $\gamma = 0.4$ | 71.8 | 67.5 | 70.7 |
| $\gamma = 0.5$ | 71.4 | 68.2 | 70.6 |
| $\tau = 0.5$ | 71.2 | 67.9 | 70.4 |
| $\tau = 0.7$ | 71.8 | 67.5 | 70.7 |
| $\tau = 0.9$ | 71.1 | 68.2 | 70.3 |

$AP_{50}$ (70.7), and both lower (0.5) and higher (0.9) thresholds result in small degradations. Overall, these results indicate that our method is not overly sensitive to threshold selection, and we use $\gamma = 0.4$ and $\tau = 0.7$ as default in all experiments.

## D. Multi-task evaluation on DIOR

We further evaluate SIKD under multi-task class-incremental settings on DIOR. Specifically, we consider three settings: 10+5+5, 5+5+5+5, and 10+2+2+2+2+2, and report the final-phase $AP_{50}$ in Table 8. Across all settings, SIKD consistently outperforms CL-DETR and CL-DETR*. Notably, the performance gap becomes larger as the number of incremental phases increases, suggesting that SIKD better mitigates forgetting and maintains effective knowledge transfer over longer learning sequences.

## E. More visualization

### E.1. t-SNE visualization

To qualitatively assess representation drift in class-incremental detection, we visualize the learned object features using t-SNE on the COCO 2017 validation set under the 70+10 setting in Figure 6. The baseline shows fragmented and less cohesive clusters, indicating unstable representations and aggravated old–new confusion after incremental updates. In contrast, our SIKD yields a more compact and well-structured embedding space that more closely resembles the joint-training reference. This suggests that the proposed symbiosis-inspired distillation better preserves old-class feature geometry while effectively incorporating new classes. Overall, this qualitative observation aligns with our quantitative results, supporting that SIKD alleviates catastrophic forgetting and improves feature consistency across incremental tasks.

### E.2. CFE module visualization

In Fig. 7, we visualize the per-detection transition in predicted class and confidence before and after applying CFE. Arrows indicate the mapping from the pre-CFE state to the post-CFE state.

## F. Efficiency of our method

Table 9 reports inference and per-image training GFLOPs and parameter counts on COCO 2017 (70+10) with input size (1064×800). All methods keep inference at 125 GFLOPs since no test-time modules are added. The baseline uses one forward of the frozen old model together with one forward–backward of the new detector, totaling 500 GFLOPs for training. Raw KD increases the training cost to 644 GFLOPs by adding an extra forward through the new decoder and the associated

*Table 8.* Experimental results ($AP_{50}$, %) on the DIOR dataset under the multi-task settings. Best results are in **bold**.

| Methods | 10+5+5 | 5+5+5+5 | 10+2+2+2+2+2 |
|---|---|---|---|
| CL-DETR | 42.9 | 39.4 | 35.2 |
| CL-DETR* | 47.1 | 37.1 | 34.1 |
| SIKD (Ours) | **66.5** | **61.4** | **61.4** |

*Table 9.* Efficiency on COCO 2017 (70+10). GFLOPs[1] denotes inference FLOPs; GFLOPs[2] denotes training FLOPs per image.

| Method | GFLOPs[1] | GFLOPs[2] | #Params (M) |
|---|---|---|---|
| baseline | 125 | 500 | 82.41 |
| Raw KD | 125 | 644 | 82.41 |
| CFE | - | 2.169 | 1.05 |
| SIKD | 125 | 694.169 | 83.46 |

KD losses. SIKD retains these costs and further adds the training-only CFE block with both forward and backward, plus one additional forward of the frozen old decoder using enhanced queries for slot-aligned supervision, reaching 694.169 GFLOPs in training. CFE has no test-time cost, so inference remains unchanged.

## G. Analysis of detection predictions

As shown in Fig. 8, we qualitatively compare detection predictions of the baseline and our SIKD on COCO 2017 under the 70+10 setting. In Fig. 8(a), SIKD prevents the old class *oven* from being confused with the visually similar new class *microwave*. In Fig. 8(b), SIKD avoids misclassifying the old class *carrot* as the new class *apple* and successfully detects hard instances that are missed by the baseline. In Fig. 8(c), SIKD reduces forgetting of the old class *orange*, whereas the baseline misses many orange objects. In Fig. 8(d) and Fig. 8(e), on images containing only old classes such as *orange* or *boat*, SIKD suppresses spurious predictions of new classes while reducing missed detections of old-class objects. These results demonstrate that SIKD simultaneously alleviates old–new confusion and preserves detection quality on old classes.

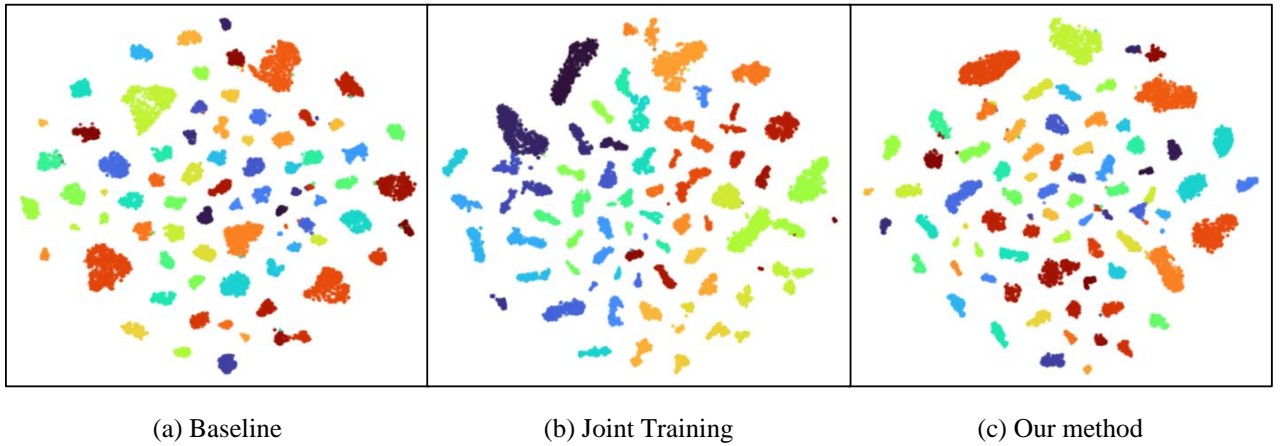

(a) Baseline        (b) Joint Training        (c) Our method

*Figure 6.* t-SNE visualization of object features on the COCO 2017 validation set. (a) Baseline after the final incremental task on COCO 2017 (70+10) setting, (b) Joint Training as an upper-bound model trained once on the union of old and new classes with full annotations (non-incremental), and (c) our method (SIKD) after the final incremental task on COCO 2017 (70+10) setting.

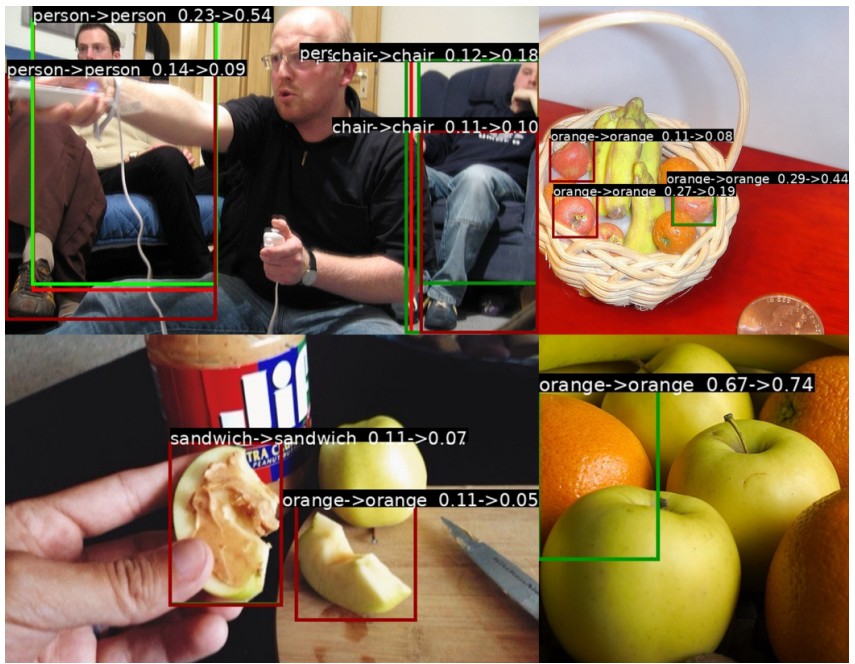

*Figure 7.* Visualization of the CFE module. For each detection, class id and confidence are shown as "old→new" with the left value before CFE and the right value after CFE.

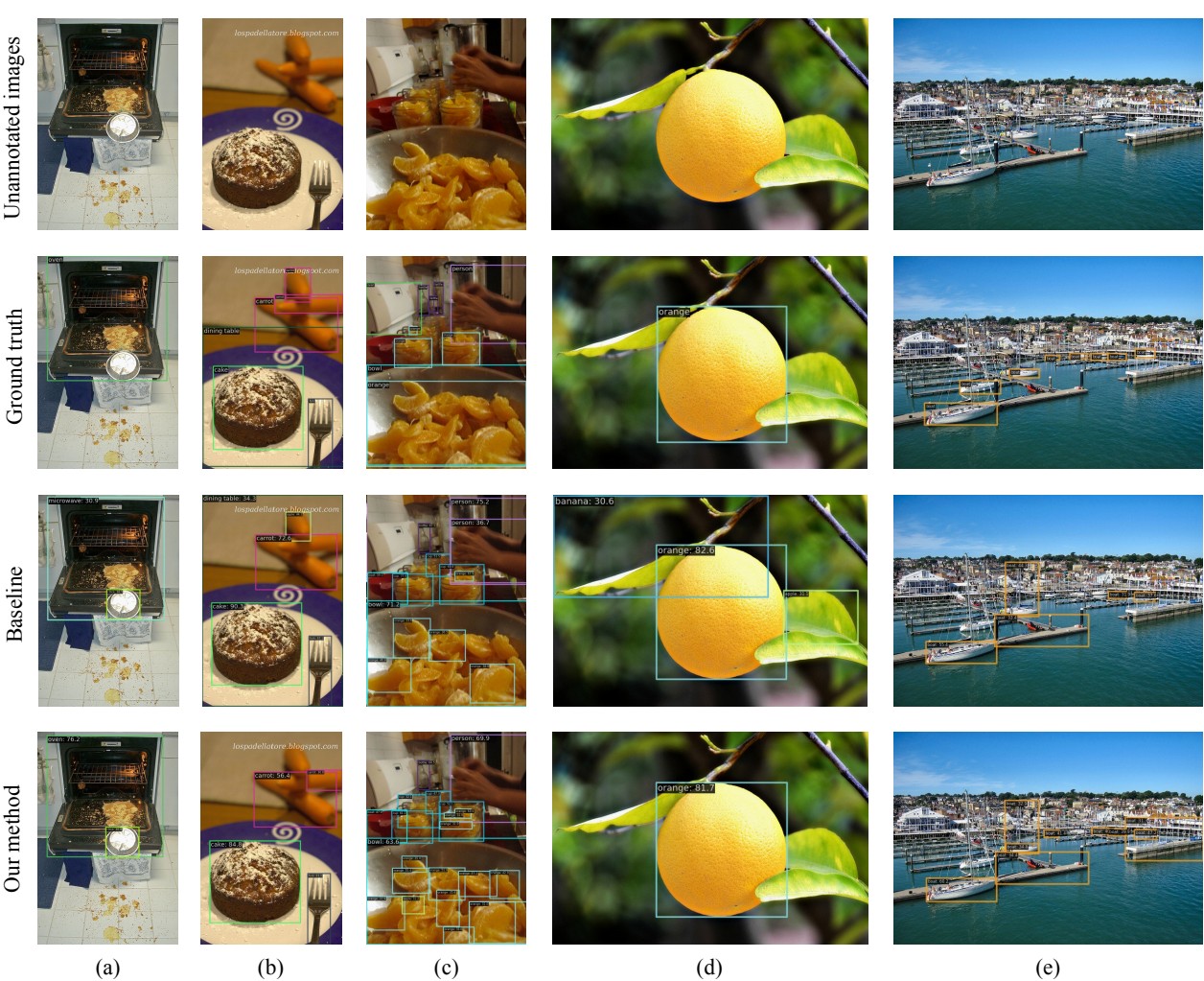

*Figure 8.* Qualitative visualization of detection predictions from the baseline and our method (SIKD) on COCO 2017 (70+10) setting. In (a), *oven* is an old class and *microwave* is a new class. In (b), *carrot* is an old class and *apple* is a new class. In (c), both *orange* and *person* are old classes. In (d), *orange* is an old class, whereas *banana* and *apple* are new classes. In (e), *boat* is an old class.

