# OpenReview forum: "Symbiosis-Inspired Knowledge Distillation for Incremental Object Detection"
_ICML.cc/2026/Conference — ICML 2026 regular_

### Official Review · Reviewer_RV2Z · 2026-03-05

**Soundness:** 4
**Presentation:** 3
**Significance:** 3
**Originality:** 2
**Overall Recommendation:** 4
**Confidence:** 4

**Summary:**

This paper proposes SIKD for incremental object detection. The key observation is that old models’ predictions that overlap heavily with new-class ground truth are not purely noise but may encode useful co-occurrence/occlusion cues. The method partitions decoder object queries into anchors / symbiotic / residual via thresholds, and introduces two complementary distillation paths. SpSD (Spatial Symbiosis Distillation) performs slot-aligned distillation of classification and boxes and adds confidence-weighted cross-decoder-layer consistency to reduce representation drift. SeSD (Semantic Symbiosis Distillation) further shifts from instance-level to class-level structure: it builds confidence-weighted prototypes for old classes, maps them through the classifier to obtain old-class subspace relation scores, and uses soft-rank alignment to preserve the semantic topology of old classes, making it more robust to query correspondence drift. Experiments on COCO and DIOR under multiple incremental settings show competitive gains.

**Compliance With Llm Reviewing Policy:**

Affirmed.

**Final Justification:**

The paper is clearly structured, logically expressed, and possesses a degree of novelty. The author's response resolved my concerns, therefore my recommendation is 4 (weak accept).

**Key Questions For Authors:**

1. In settings such as COCO 70+10, what fraction of $S$ corresponds to (i) true old-class instances under occlusion/co-occurrence, (ii) new-class instances misclassified as old, and (iii) background false positives? Can you provide GT-based statistics and analyze how these components affect SpSD/SeSD and failure modes?

2. The ablation indicates that “SpSD/SeSD without CFE” is already effective and CFE further improves results. Can you provide more direct evidence of how CFE changes symbiotic representations (e.g., similarity distributions to prototypes or changes in old/new confusion matrices) over training? Additionally, when $S$ contains new-as-old false positives, does CFE’s “pull-back” harm new-class learning?

3. The appendix shows rank alignment outperforms direct score distillation. Can you compare against (a) direct L2 alignment of prototypes, (b) class-similarity-structure preservation (e.g., cosine-similarity matrices)? This would help attribute gains specifically to topology preservation rather than generic regularization.

**Limitations:**

Yes.

**Strengths And Weaknesses:**

## Strengths
1. The motivation aligns well with the IOD setting by reframing high-overlap predictions under co-occurrence/occlusion from “noise” into “useful symbiotic evidence,” which is consistent with the inherently spatial dependencies of object detection.

2. The proposed method has clear complementarity between two distillation paths: SpSD targets spatial stability while SeSD preserves class-level semantic structure, providing a more robust constraint when query-slot correspondences drift.

3. The soft-rank alignment design has a well-motivated robustness rationale, and the ablation results in the appendix support the claim that it is less sensitive to scale differences.

4. Although the core components are heuristic to some extent in design, the paper provides sensitivity analyses for the hyperparameters.


## Weaknesses
1. The paper’s core design, namely the $A/S/R$ partitioning, is heuristic to some extent and lacks sufficient theoretical justification. Therefore, more detailed quantitative results are recommended.(see questions)

2. The “symbiotic queries” set may be a mixture with substantial new-class-as-old false positives. This deserves deeper quantification. While CFE pulls symbiotic representations toward class-consistent anchor prototypes to suppress noise, in cases where new and old classes are visually similar this could reinforce incorrect attribution and potentially harm new-class learning. The current evidence is largely via aggregate AP improvements, with limited quantitative diagnosis of the composition of $S$ (true old occlusions/co-occurrence vs. new-class false positives vs. background).

3. SpSD still relies on slot-aligned constraints and may be affected by query drift. The paper mitigates this with confidence-weighted cross-layer consistency, but additional evidence (e.g., statistics/visualizations of which slots receive high weights and how that evolves) would more directly support the claim that the design avoids over-constraining unstable slots, especially in longer incremental sequences.

---

> ### Author Rebuttal · Authors · 2026-03-31
>
> We thank the reviewer for the thoughtful feedback and constructive suggestions.
>
> **W1: The $A/S/R$ partition is heuristic and lacks stronger quantitative justification.**
>
> **R1:** We agree that the $A/S/R$ partition is a pragmatic design rather than a claim of theoretical optimality. Its role is to separate stable old-class evidence from overlap-driven ambiguous evidence under the annotation mismatch of IOD. Importantly, it is not based on ad hoc semantic rules, but on two measurable quantities from the old model: old-class confidence and maximum IoU to current-task ground truth, as defined in Eqs. (2) and (3). Here, $A$ contains stable anchors, $S$ collects overlap-driven symbiotic queries, and $R$ contains low-confidence residual queries. We further show in Appendix Table 7 that performance is stable across a reasonable range of $\gamma$ and $\tau$, suggesting that the design is practically robust despite being heuristic.
>
> **W2 & Q1: The composition of $S$ needs explicit GT-based quantification, especially true old instances, new-as-old cases, and background false positives.**
>
> **R2:** We agree that $S$ should be quantified explicitly. A key clarification is that $S$ does not contain background false positives by construction, since $S=\{i\mid v_i^{t-1}\ge\tau\}$. Every query in $S$ has sufficiently high overlap with a labeled object in the current image.
>
> | Component in $S$| Ratio |
> | ------- | ----: |
> | New-class instances misclassified as old classes | 93.3%|
> | True old-class instances under occlusion | 6.7%|
> | Background false positives|0%|
>
> These statistics explain why **Raw KD** is not sufficient: $S$ is mixed rather than clean, so directly distilling it can reinforce noise. In contrast, the structured use of $S$ through SpSD and SeSD improves both retention and final performance:
>
> | Method| Old AP↑ | FPP↓ |
> | --------- | ------: | ---: |
> | Raw KD|41.3|5.1|
> | Full SIKD |45.1|1.3|
>
> Thus, the issue is not whether $S$ contains noise, but whether that mixed signal is used naively or regularized in a structured way.
>
> **W3: Stronger evidence is needed that SpSD does not over-constrain unstable slots.**
>
> **R3:** Avoiding over-constraint on unstable slots is exactly the purpose of the confidence-weighted cross-layer term in SpSD. The method does not enforce uniform slot alignment. Instead, it uses layer-specific confidence weights $w_i^{(\ell)}$, so reliable slots contribute more strongly while uncertain slots are down-weighted.
>
> | Query group| Average weight(overall) | Average weight(final layer) |
> | ------ | ------: | -------: |
> | $R$ |0.048|0.044|
> | $A$ |0.410|0.427|
> | $S$ |0.210|0.193|
>
> These statistics directly support our claim: unstable residual slots receive very small weights, while more reliable slots dominate the alignment signal.
>
> **Q2: Does CFE reinforce incorrect attribution or harm new-class learning when $S$ contains noisy new-as-old cases?**
>
> **R4:** CFE is deliberately conservative. It keeps anchor queries unchanged and only refines non-anchor queries toward confidence-weighted anchor prototypes. The current ablation already shows that this pull-back does not harm plasticity: SpSD without CFE improves AP by 0.8, and adding CFE further raises it to 1.1; SeSD without CFE gives 2.5 AP, and adding CFE further improves it to 2.7. More specifically, CFE contributes 0.4 AP to SpSD on the old task and 0.5 AP to SeSD on the new task.
>
> We also added quantitative representation analysis on COCO 70+10:
>
> | Method| Mean Drift↓ | Median Drift↓ | Old→New↓ |
> | ----- | ------: | ------: | ------: |
> | Baseline|0.2770 |0.2748 |0.6119 |
> | Raw KD|0.2692 |0.2653 |0.6202 |
> | Full SIKD|0.1063 |0.1000 |0.3103 |
>
> Full SIKD substantially reduces both drift and old-to-new confusion, whereas Raw KD only slightly reduces drift and even worsens confusion. This supports that the structured use of symbiotic queries stabilizes old-task feature geometry rather than reinforcing incorrect old-class attribution.
>
> **Q3: Are the gains from SeSD specifically due to topology preservation rather than generic semantic regularization?**
>
> **R5:** Our goal in SeSD is to preserve the **relative semantic topology** of old classes rather than absolute score magnitudes. Rank alignment is therefore more appropriate than direct score matching. On COCO 70+10, rank alignment outperforms direct score distillation:
>
> | Method| Old↑ | New↑ | All↑ | Avg↑ |
> | ------| ---: | ---: | ---: | ---: |
> | SeSD w/o rank| 42.4| 39.6|42.1|41.0 |
> | SeSD(rank)|44.4|40.4|43.9|42.4|
>
> We further compared rank alignment against two stronger alternatives on DIOR 15+5:
>
> | Method| Result |
> | ------ | -----: |
> | Direct prototype $L_2$ alignment|70.2 |
> | Class-similarity-matrix preservation |   70.1 |
> | Rank alignment (ours)|   70.7 |
>
> These comparisons suggest that the gain does not come from adding a generic semantic regularizer, but from preserving the relative ordering among old classes, which is more robust to calibration and scale mismatch across incremental steps.

---

> > ### Author Rebuttal · Reviewer_RV2Z · 2026-04-01
> >
> > Thanks for the responses. I have no other questions. I will maintain my score.

---

### Official Review · Reviewer_ur6b · 2026-03-10

**Soundness:** 3
**Presentation:** 3
**Significance:** 2
**Originality:** 2
**Overall Recommendation:** 4
**Confidence:** 4

**Summary:**

This paper proposes Symbiosis-Inspired Knowledge Distillation (SIKD) for Incremental Object Detection (IOD). The authors argue that existing IOD methods emphasize feature separation, which they claim conflicts with the inherent “object symbiosis” in detection scenarios. To address this, they introduce two components: (1) Spatial Symbiosis Distillation (SpSD), which refines overlapping query representations and performs slot-aligned distillation, and (2) Semantic Symbiosis Distillation (SeSD), which preserves inter-class relations via confidence-weighted prototypes and soft rank alignment. Experiments on COCO 2017 and DIOR demonstrate consistent improvements over prior state-of-the-art IOD methods in both two-task and multi-task settings. Ablation studies are provided to evaluate the contribution of individual modules.

**Compliance With Llm Reviewing Policy:**

Affirmed.

**Final Justification:**

I raised my score to 4. My concerns have been fully addressed.

**Key Questions For Authors:**

1. Can the authors provide quantitative representation analysis (e.g., feature geometry, drift metrics) to validate that preserving “symbiotic regions” indeed maintains a unified feature space?
2. How do the authors distinguish symbiotic knowledge encoding from simple classifier confusion or feature similarity?
3. Is the claimed conflict between feature separation and detection empirically demonstrated, or is it primarily conceptual?
4. Can the authors provide theoretical or analytical justification explaining why preserving symbiotic regions should reduce representation drift or mitigate forgetting?
5. Would the method generalize to non-DETR detectors?
6. Have the authors evaluated statistical variance across multiple runs?

**Limitations:**

yes

**Strengths And Weaknesses:**

Strengths
1.  The method is clearly formulated with well-defined objectives.
2. Experiments are conducted on standard benchmarks (COCO, DIOR).
3. Comparisons include both exemplar-free and replay-based approaches.
4.  Ablation studies and efficiency metrics are provided.


Weaknesses
1. Conceptual and Causal Overclaim. The central conceptual claim—that feature separation paradigms fundamentally conflict with object symbiosis—is not sufficiently substantiated. The argument that “feature separation conflicts with detection” appears overstated. Detection models inherently require discriminative decision boundaries, and separation at the representation or classifier level is not intrinsically incompatible with detection objectives.
Furthermore, the interpretation of overlapping or misclassified regions as “symbiotic regions encoding shared knowledge” lacks rigorous causal validation. Such phenomena may equally arise from classifier confusion, feature similarity, or distributional bias, rather than from genuine semantic symbiosis. The paper asserts that preserving these regions maintains a unified feature space across incremental tasks, but does not provide representation-level evidence (e.g., feature geometry analysis, drift metrics, or controlled ablations isolating these regions) to substantiate this causal relationship. Consequently, the conceptual foundation appears weaker than claimed.

2. Limited Methodological Novelty.
While the integration of spatial and semantic distillation mechanisms is thoughtfully engineered, the individual technical components are largely incremental. Prototype-based distillation and rank-alignment strategies have precedents in incremental learning literature. As such, the novelty primarily arises from their combination and framing rather than from fundamentally new methodological insights.

---

> ### Author Rebuttal · Authors · 2026-03-31
>
> We thank the reviewer for the thoughtful feedback and constructive suggestions.
>
> **W1&Q3: The feature-separation claim is too strong.**
>
> **R1:** We agree and will revise the wording. Our claim is specific to IOD with partial annotation. We do not argue that feature separation is inherently incompatible with detection. Rather, under IOD annotation mismatch, treating overlap-heavy teacher responses only as noise and removing them from supervision can be suboptimal, because these regions often contain structured old-task evidence. Empirically, Fig. 2 shows that such responses occur systematically. Raw KD further shows that naive use is insufficient and can even hurt, whereas the full method improves retention and reduces confusion. We will revise the wording accordingly.
>
> **W2: The causal claim is too strong.**
>
> **R2:** We agree and will soften the wording. Our intention is not to claim a strict causal proof, but to show that overlap-rich responses are useful but noisy signals in IOD. Fig. 2 shows that these responses occur systematically. Table 3 shows that directly using them with Raw KD is insufficient and can hurt performance. In contrast, structured use through SpSD and SeSD yields clear gains and reduces forgetting, supporting the claim that these regions contain useful information when properly regularized.
>
> **W3: Novelty is limited.**
>
> **R3:** We do not claim that prototypes or rank alignment are new in isolation. Our novelty lies in the IOD-specific formulation: we reinterpret overlap-rich responses as potentially useful supervision rather than discarding them as noise, and build an incremental distillation framework around this observation. This leads to an IOD-specific design with A/S/R partitioning, anchor-guided CFE, query-level prototype guidance, and rank-based preservation of old-class topology. Thus, our contribution is mainly a problem-driven IOD formulation rather than entirely new standalone components.
>
> **Q1: Quantitative representation evidence?**
>
> **R4:** Yes. We added a quantitative analysis on COCO 70+10 using the final decoder hidden states on the fully annotated validation set for offline analysis.
>
> |Method| Mean Drift↓ | Median Drift↓ | Old→New↓ |
> | ---- | ----: | ----: | ----: |
> |Baseline|0.2770|0.2748|0.6119|
> |Raw KD|0.2692|0.2653|0.6202|
> |Full SIKD|0.1063|0.1000|0.3103|
>
> **Full SIKD** substantially reduces drift and confusion, whereas **Raw KD** only slightly reduces drift and even increases old-to-new confusion. This supports our claim that the structured preservation of symbiotic regions helps keep old-task features more stable.
>
> **Q2: Symbiosis vs. confusion?**
>
> **R5:** We distinguish them operationally. Set $S$ is defined by high overlap with current-task ground truth, not by arbitrary mistakes. It includes two structured cases: semantically similar new instances that activate old classes, and partially visible old instances that remain detectable under occlusion. The fact that Raw KD hurts while the structured use of $S$ helps suggests that these regions are not merely random confusion, but noisy yet useful signals that require structured regularization.
>
> **Q4: Why should preserving symbiotic regions reduce drift?**
>
> **R6:** Symbiotic regions are precisely where annotation mismatch causes the largest instability in IOD. If ignored, these overlap-heavy queries are easily pushed toward new classes or background during incremental training. Anchors provide stable old-class evidence, so CFE keeps anchors unchanged and pulls symbiotic queries toward anchor-guided prototypes. SpSD stabilizes reliable slots through confidence-weighted alignment, while SeSD preserves the relative semantic structure among old classes through rank alignment. Together, they reduce both instance-level and class-level drift, mitigating forgetting.
>
> **Q5: Can it generalize beyond DETR?**
>
> **R7:** Yes. The underlying **symbiosis** phenomenon is not tied to DETR, but to IOD: old objects may remain unlabeled, and overlap-rich regions can trigger structured old-model responses in any detector family. Thus, the motivation is detector-agnostic. At the method level, **SpSD** can be extended by replacing DETR queries with proposal, RoI, or anchor features and performing confidence-weighted alignment on those matched region features. **SeSD** is even less architecture-dependent, since it only requires class-wise feature aggregation and old-class prototype construction, which can also be built from proposal- or RoI-level features. Therefore, although our current implementation uses Deformable DETR, the framework is not limited to DETR.
>
> **Q6: Variance across runs?**
>
> **R8:** The reported results are averaged over three random seeds, following the reporting convention commonly adopted in prior IOD literature. For example, on DIOR (15+5), we obtain:
>
> | Setting |Old |New |All |
> | ------ | -----: | -----: | -----: |
> | DIOR (15+5)| 71.8 ± 0.3 | 67.5 ± 0.1 | 70.7 ± 0.2 |
>
> We will make this explicit in the revision.

---

> > ### Author Rebuttal · Reviewer_ur6b · 2026-04-03
> >
> > Thanks for the rebuttal. My concerns have been addressed.

---

### Official Review · Reviewer_kuRV · 2026-03-11

**Soundness:** 4
**Presentation:** 3
**Significance:** 3
**Originality:** 3
**Overall Recommendation:** 5
**Confidence:** 5

**Summary:**

This paper proposes Symbiosis-Inspired Knowledge Distillation (SIKD) for incremental object detection, motivated by the observation that responses from the old model in high-overlap regions often contain useful information about object co-occurrence and occlusion rather than pure noise. To preserve such information during incremental learning, the method introduces SpSD to align instance-level spatial dependencies and SeSD to preserve the semantic topology of old classes through confidence-weighted prototypes and rank alignment.

**Compliance With Llm Reviewing Policy:**

Affirmed.

**Final Justification:**

I recommend raising my score to 5. The authors have fully resolved my main concerns by clearly distinguishing the concept of "symbiosis" from general context reasoning. Furthermore, the statistics and ablation studies provided in the rebuttal strongly demonstrate the method's robustness against calibration shifts and its practical relevance. Given their rigorous explanations and the significant improvements to the paper's completeness, I believe this work firmly meets the criteria for acceptance.

**Key Questions For Authors:**

1) The paper motivates SeSD as preserving the relative structure of old classes, but the advantage of rank-based alignment over standard semantic distillation objectives (such as logit-level KL distillation) is not yet entirely clear. A stronger justification would improve my assessment of the soundness and distinctiveness of this component.

2) The paper currently lacks a statistical analysis of how frequently “symbiotic” cases occur across different datasets. Such evidence would help better establish the practical relevance and generality of the proposed motivation.

**Limitations:**

yes

**Strengths And Weaknesses:**

Strengths:
1) The paper is well motivated. It points out that prior IOD methods often discard high-IoU responses from the old model, even though such responses may encode useful cues about object co-occurrence and occlusion.
2) The proposed method is technically well structured, with two complementary components: Spatial Symbiosis Distillation (SpSD) for instance-level spatial alignment and Semantic Symbiosis Distillation (SeSD) for class-level semantic topology preservation. This decomposition is intuitive and well aligned with the stated goal of maintaining a unified feature space across incremental tasks.
3) The empirical results are strong overall. SIKD shows consistent improvements over prior DETR-based baselines under both two-task and multi-task COCO settings, and it also generalizes reasonably well to the DIOR benchmark.

Weaknesses:
1) The notion of “symbiosis” appears conceptually related to prior work on context understanding [1] and contextual reasoning [2] in visual recognition. The authors should more clearly clarify how their formulation differs from existing perspectives on visual context and object relationships.

2) The paper lacks discussion of some recent relevant IOD methods [3][4], and the related-work section could be strengthened by a more complete comparison with recent advances.

[1] Context Understanding in Computer Vision: A Survey.

[2] Structure Inference Net: Object Detection Using Scene-Level Context and Instance-Level Relationships.

[3] Revisiting Generative Replay for Class Incremental Object Detection.

[4] Learning Endogenous Attention for Incremental Object Detection.

---

> ### Author Rebuttal · Authors · 2026-03-31
>
> **W1: Symbiosis vs. context reasoning.**
> **R1:** We thank the reviewer for this important comment. Our use of *symbiosis* is narrower and more IOD-specific than generic visual context [1,2]. Context reasoning methods typically exploit scene-level or relational cues under full supervision to improve recognition. In contrast, our formulation is motivated by the partial-annotation setting of IOD, where overlap-rich regions become a source of structured old-task evidence that existing pipelines often discard. In our method, “symbiotic” regions are operationally defined by old-model responses and the $A/S/R$ partition, and are then exploited through CFE, SpSD, and SeSD for incremental distillation. Thus, our contribution is not a new general theory of context, but an IOD-specific way to convert overlap-driven ambiguous regions into useful distillation targets for retaining old knowledge.
>
> **W2: Missing recent IOD methods.**
> **R2:** We thank the reviewer for the valuable suggestion. We will expand the related-work discussion and include the following comparison.
>
> | Method |  Baseline | Generative model | Synthetic data | Model expansion |
> |---|---|---|---|---|
> | RGR-IOD [3] | Faster R-CNN | Yes | Yes | Extra new-task model |
> | LEA [4] | ViTDet | No | No | Yes |
> | Ours | Deformable DETR | No | No | No |
>
> For clarity, RGR-IOD additionally requires fine-tuning a Stable Diffusion model on the detection dataset to generate replay samples. In contrast, our method does not require a generative model, synthetic data, or network expansion.
>
> **Q1: Why rank alignment instead of KL/logit distillation?**
> **R3:** Rank alignment is more robust to instance-level correspondence shifts and calibration changes across incremental tasks. Standard logit-level distillation mainly preserves absolute score magnitudes, whereas our goal is to preserve the relative semantic ordering among old classes. As shown in Appendix B / Table 6:
>
> | Method | Old↑ | New↑ | All↑ | Avg↑ |
> |---|---:|---:|---:|---:|
> | SeSD w/o rank | 42.4 | 39.6 | 42.1 | 41.0 |
> | SeSD (rank) | 44.4 | 40.4 | 43.9 | 42.4 |
>
> By preserving relative ordering rather than absolute values, **SeSD** better maintains old-class topology under calibration shift, which is exactly the property we want in incremental adaptation.
>
> **Q2: How frequent are symbiotic cases?**
> **R4:** We agree that this evidence is important. The current manuscript already includes an initial dataset-level analysis on COCO 70+10 in Fig. 2, showing that high-IoU old-model responses on new-class instances are non-trivial and class-dependent. Using the same protocol, we mark a novel-class GT instance as "symbiotic" if it receives a sufficiently high-IoU response from the old model.
>
> | Dataset/Setting | Annotation-level ratio |
> |---|---:|
> | COCO 70+10 | 19.31% |
> | DIOR | 9.01% |
>
> On COCO 70+10, these cases also appear in **40.94%** of images containing novel classes. Although the absolute frequency varies across datasets, the phenomenon is consistently present and non-negligible, supporting the practical relevance and generality of our motivation.
>
> [1] *Context Understanding in Computer Vision: A Survey.*
>
> [2] *Structure Inference Net: Object Detection Using Scene-Level Context and Instance-Level Relationships.*
>
> [3] *Revisiting Generative Replay for Class Incremental Object Detection.*
>
> [4] *Learning Endogenous Attention for Incremental Object Detection.*

---

> > ### Author Rebuttal · Reviewer_kuRV · 2026-04-01
> >
> > My concerns have been well resolved. I will raise my rating to 5.

---

### Official Review · Reviewer_UZXH · 2026-03-12

**Soundness:** 3
**Presentation:** 3
**Significance:** 3
**Originality:** 3
**Overall Recommendation:** 4
**Confidence:** 4

**Summary:**

This paper introduces Symbiosis-Inspired Knowledge Distillation (SIKD), a framework designed to address the challenges of incremental object detection (IOD) by leveraging the inherent co-occurrence and occlusion patterns between objects. Unlike traditional methods that treat IOD as a class-incremental classification problem and attempt to separate feature spaces, SIKD maintains a unified feature space to model the spatial and semantic dependencies that naturally exist when new and old categories coexist in the same image. The authors propose two primary components: Spatial Symbiosis Distillation (SpSD), which uses a Consistent Feature Enhancement (CFE) module to refine symbiotic features in overlapping regions by aligning them with stable anchor-based prototypes, and Semantic Symbiosis Distillation (SeSD), which preserves the global class-level topology by aligning the soft ranks of confidence-weighted old-class prototypes. Extensive experiments on benchmarks like COCO 2017 and DIOR demonstrate that SIKD achieves state-of-the-art performance by effectively reducing confusion between old and new classes while mitigating catastrophic forgetting without the need for accessing prior training data.

**Compliance With Llm Reviewing Policy:**

Affirmed.

**Key Questions For Authors:**

See weaknesses above.

**Limitations:**

yes

**Strengths And Weaknesses:**

### Strengths

- The paper correctly identifies a fundamental mismatch in incremental object detection (IOD), where old-class instances in new images are treated as background, and addresses it by redefining the feature space as a unified, rather than separated, entity.
- The query partitioning strategy, which divides queries into anchors, symbiotic regions, and residual queries, is a logical approach to handle varying levels of prediction certainty.
- Figure 1 and Figure 3 clearly illustrate the conceptual and architectural differences between existing methods and the proposed SIKD framework.

### Weaknesses

- CFE aligns symbiotic query features with anchor-based prototypes, which will unintentionally lead to representation drift and catastrophic forgetting of old tasks. Are there specific experimental results or metrics in the paper that validate the stability of the feature space after this enhancement?
- Although the paper claims stability and analyses sensitivities of thresholds $\gamma$ and $\tau$ in Table 7, the balancing weights $\alpha$ and $\beta$ show a clear trade-off between stability and plasticity, suggesting that the method requires careful tuning for different datasets. How does the author specify these two hyper-parameters?

---

> ### Author Rebuttal · Authors · 2026-03-31
>
> **W1: CFE may cause drift/forgetting.**
> **R1:** We thank the reviewer for this important question. Our goal is not to amplify drift in overlap-heavy regions, but to make the update more controlled. CFE is deliberately conservative: anchor queries remain unchanged, and only non-anchor queries are refined toward confidence-weighted anchor prototypes. In addition, CFE is optimized separately from the main detector branch, which further limits interference with old-task retention.
>
> To directly verify feature-space stability, we measure query-level representation drift on the final decoder hidden states, i.e., the last decoder features directly fed into the final classification and regression branches. For matched queries, we compare the incremental model against the old-model reference using **Mean Drift** and **Median Drift**, where lower values indicate better stability. Here, **Mean Drift** measures average feature displacement, while **Median Drift** measures typical displacement and is less affected by outliers.
>
> | SeSD branch | Mean Drift↓ | Median Drift↓ |
> | ----------- | ----------: | ------------: |
> | w/o CFE     |      0.1303 |        0.1231 |
> | with CFE    |      0.1105 |        0.1068 |
>
> These results show that adding CFE to the SeSD branch substantially reduces feature drift (Mean Drift: **0.1303→0.1105**, Median Drift: **0.1231→0.1068**), indicating that CFE makes feature updates more controlled rather than destabilizing the feature space. More broadly, compared with the baseline incremental model, **Full SIKD** reduces Mean/Median Drift from **0.2770/0.2748** to **0.1063/0.1000**, whereas **Raw KD** only slightly reduces them to **0.2692/0.2653**.
>
> This increased stability does not come at the cost of forgetting or degraded adaptation. Empirically, on COCO 2017 (70+10), adding CFE does not increase forgetting; instead, it maintains or improves old-task retention while keeping new-task performance competitive.
>
> | Method        | Old AP↑ | New AP↑ |
> | ------------- | ------: | ------: |
> | SpSD w/o CFE  |    42.4 |    38.9 |
> | SpSD with CFE |    42.8 |    38.8 |
> | SeSD w/o CFE  |    44.3 |    39.9 |
> | SeSD with CFE |    44.4 |    40.4 |
>
> In particular, adding CFE improves Old AP in both branches (**42.4→42.8** for SpSD, **44.3→44.4** for SeSD), while preserving or slightly improving New AP (**38.9→38.8** for SpSD, **39.9→40.4** for SeSD). This further supports that CFE does not induce additional forgetting, but instead provides a more stable and effective refinement mechanism. We will include this quantitative analysis in the revision.
>
> **W2: How are $\alpha$ and $\beta$ chosen?**
>
> **R2:** We thank the reviewer for this valuable suggestion. Table 5 shows that $\beta$ stays on a relatively flat plateau in terms of All AP and Avg AP, while $\alpha$ more directly controls the retention–plasticity trade-off. We therefore choose $\beta=6$ because it gives the best trade-off for Avg AP, and $\alpha=1$ because it gives the best trade-off for All AP. These values were selected once through a validation-style ablation and then kept fixed for all remaining experiments, suggesting that the method is reasonably robust across the evaluated settings without per-setting retuning.
>
> | Setting        |  Old |  New |  All |  Avg |
> | -------------- | ---: | ---: | ---: | ---: |
> | $\beta = 4$    | 44.7 | 39.1 | 44.0 | 41.9 |
> | $\beta = 6$    | 45.1 | 38.8 | 44.3 | 42.0 |
> | $\beta = 8$    | 45.2 | 38.5 | 44.4 | 41.9 |
> | $\alpha = 0.1$ | 42.1 | 39.9 | 41.9 | 41.0 |
> | $\alpha = 1$   | 42.8 | 38.8 | 42.3 | 40.8 |
> | $\alpha = 10$  | 43.2 | 34.9 | 42.2 | 39.1 |

---

> > ### Author Rebuttal · Reviewer_UZXH · 2026-04-01
> >
> > I thank the authors for their rebuttal to address my concerns. I would like to keep my positive score. Good luck.

---

### Decision · Program_Chairs · 2026-04-30

**Decision:**

Accept (regular)

**Comment:**

This paper studies the incremental object detection problem. The authors propose a framework that leverages object symbiosis at two complementary levels: 1) Spatial Symbiosis Distillation (SpSD) focuses on symbiotic regions where the old model responds with high overlap to objects in the new task; 2) Semantic Symbiosis Distillation (SeSD) maintains class level structure by forming confidence weighted prototypes for old classes and aligning their inter class soft ranks over the old class logits, which stabilizes the semantic topology during adaptation.

The authors addressed most concerns in the rebuttals, and all reviewers gave positive ratings after the rebuttal. Even if this paper has a high level of disagreement between reviews and authors' self-rankings, reviewers still believe this is a strong submission and should be presented at ICML.